# Identifying degradation patterns of lithium ion batteries from impedance spectroscopy using machine learning

Yunwei Zhang [1,2,6], Qiaochu Tang[2,3,4,6], Yao Zhang [5], Jiabin Wang[2,3,4], Ulrich Stimming[2,3,4,7 ✉] & Alpha A. Lee[1,2,7 ✉]

Forecasting the state of health and remaining useful life of Li-ion batteries is an unsolved challenge that limits technologies such as consumer electronics and electric vehicles. Here, we build an accurate battery forecasting system by combining electrochemical impedance spectroscopy (EIS)—a real-time, non-invasive and information-rich measurement that is hitherto underused in battery diagnosis—with Gaussian process machine learning. Over 20,000 EIS spectra of commercial Li-ion batteries are collected at different states of health, states of charge and temperatures—the largest dataset to our knowledge of its kind. Our Gaussian process model takes the entire spectrum as input, without further feature engineering, and automatically determines which spectral features predict degradation. Our model accurately predicts the remaining useful life, even without complete knowledge of past operating conditions of the battery. Our results demonstrate the value of EIS signals in battery management systems.

[1] Cavendish Laboratory, University of Cambridge, Cambridge CB3 0HE, UK. [2] The Faraday Institution, Quad One, Becquerel Avenue, Harwell Campus, Didcot OX11 0RA, UK. [3] Chemistry – School of Natural and Environmental Sciences, Newcastle University, NE1 7RU Newcastle upon Tyne, UK. [4] North East Centre of Energy Materials (NECEM), Newcastle University, NE1 7RU Newcastle upon Tyne, UK. [5] Department of Applied Mathematics and Theoretical Physics, University of Cambridge, Cambridge CB3 0WA, UK. [6] These authors contributed equally: Yunwei Zhang, Qiaochu Tang. [7] These authors jointly supervised this work: Ulrich Stimming, Alpha A. Lee. ✉email: Ulrich.Stimming@newcastle.ac.uk; aal44@cam.ac.uk

Li-ion batteries enable a wide variety of technologies that are integral to modern life by virtue of their high energy and power density[1–4]. However, a key stumbling block to advancing those technologies is the unpredictability of battery degradation: accurate prediction of battery state of health (SoH) and remaining useful life (RUL) is needed to inform the user whether a battery should be replaced and avoid unexpected capacity fade. Moreover, battery prognosis is crucial to expanding the recycling sector, enabling facilities to decide whether a battery should be recycled as scrap metal or used for less demanding "second-life" applications.

The conventional approach to battery forecasting relies on modelling microscopic degradation mechanisms, such as the growth of the solid-electrolyte interphase[5,6], lithium plating[7,8] and active material loss[9,10]. Although offering physical insights, characterising and simulating every degradation mechanism is unscalable. To overcome this challenge, recent literature focuses on data-driven approaches[11,12]. The idea is to perform real-time, non-invasive measurements on the battery, and use statistical machine learning to relate those measurements to battery health without modelling a physical mechanism. However, the challenge of data-driven approaches is defining a set of physically informative inputs, and building a robust statistical model.

Features derived from the charging and discharging curve are by far the most commonly used inputs because typical battery management systems collect current–voltage data[13–18]. Compared with the usual current–voltage data, electrochemical impedance spectroscopy (EIS), which obtains the impedance over a wide range of frequencies by measuring the current response to a voltage perturbation or vice versa[19–21], is known to contain rich information on all materials properties, interfacial phenomena and electrochemical reactions. This directly relates to possible degradation inside the battery and is able to track the status of the battery[22]. However, deploying EIS to predictive battery diagnosis is hampered by the high dimensionality of the spectrum—EIS records the real and imaginary part of the impedance over a frequency range that spans multiple decades. Although qualitative changes are apparent, it is challenging to pick out quantitative features correlated with degradation. Existing approaches reduce the spectrum into lower dimensional features: the spectrum is either interpreted by fitting to an equivalent circuit model[19,22–28] (recent work employed machine learning to aid the fit[29])—the fit is often non-unique and it is questionable whether a purely electrical model can capture the physical, chemical and materials properties and processes of a battery—or focusing only on handpicked frequencies[30–32].

Recent advances in machine learning show that one can feed the entire dataset as input into the model without handpicking features, and let the model select the most relevant variables. Those models have been developed for degradation diagnosis, such as using a Gaussian process model to predict the future capacity[33,34] and state of charge (SoC)[17], and using a regularised linear model to predict cycle life[18]. However, those models are all developed with the charging and discharging curve as input. The power of any model is circumscribed by the information content of the inputs, and forecasting the late-stage behaviour of batteries with data from early life—the most relevant problem—is still a significant challenge.

In this paper, we show that gaussian process regression (GPR) can accurately estimate the capacity and predict RUL using the EIS spectrum, which are key indicators of the SoH of a battery. We generate the largest dataset, to our knowledge, of EIS measurements of commercial Li-ion batteries (LCO/graphite) over a wide range of frequencies at different temperatures and SoC, totalling over 20,000 EIS spectra. Moreover, our method can estimate the capacity and RUL of batteries cycled at three constant temperatures, at any point of its life, from a single impedance measurement. Our model is more accurate than conventional methods, which use features of the discharging curve, and our results can be attributed back to the impedance spectrum, providing information on which frequencies are the most salient.

## Results

**Capacity estimation.** We first consider a setting where the user wants to estimate the capacity of a battery using the EIS of the current cycle, with the knowledge of the temperature, which is kept constant throughout, and the SoC (state I–IX shown in Supplementary Fig. 1).

We train the EIS-Capacity GPR model on four cells cycled at room temperature of 25 °C (marked as 25C01–25C04), and test it on the other four cells (marked as 25C05–25C08). Figure 1 shows that the model accurately estimates the capacity of the testing cells. Figure 1a shows the result of 25C05 cell for the state V (15 min resting after fully charging); the results at other states are similarly positive and shown in Supplementary Fig. 2. Out of all the states of I–IX, the model is most accurate at electrochemically stable states (i.e. the state V/IX, which is fully charged/discharged after resting), where electrochemical measurements on cells are more consistent. Figure 1b shows the measured capacity against the estimated capacity of all four testing cells. We note that all

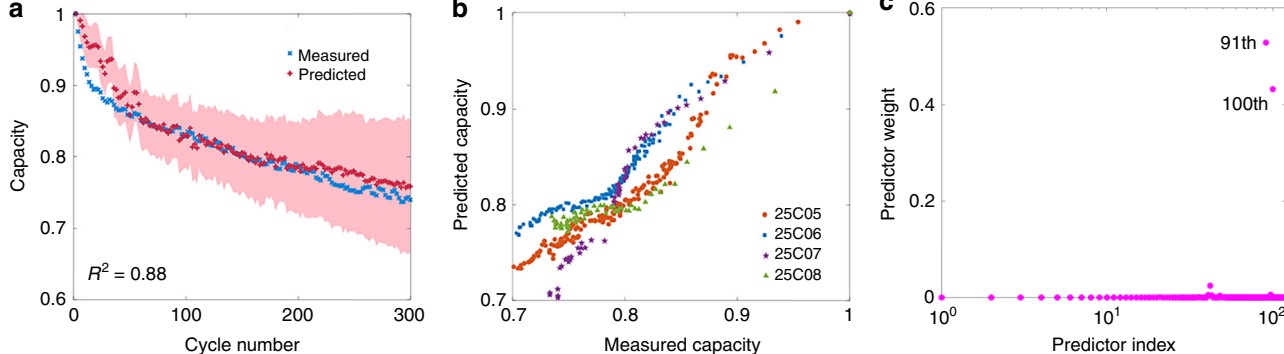

**Fig. 1 Estimating battery capacity. a** Estimated (red curve) and measured (blue curve) capacity as a function of cycle number for the 25C05 cell. The coefficient of determination ($R^2$) of this model is shown on the left bottom. **b** The measured capacity against the estimated capacity of all four testing cells cycled at 25 °C. The capacity is normalised against the starting capacity in each case. **c** ARD shows that the impedance at low frequency is most correlated with degradation. The pink points correspond to the 120 frequencies in the range of 0.02 Hz–20 kHz. The GPR model assigns the largest weights to the 91st and 100th features, corresponding to 17.80 and 2.16 Hz, respectively. The less relevant features have weights close to zero.

testing cells are charged and discharged the same way as the training cells; the ability of our model to estimate the cells cycled at different operating charge/discharge rates needs to be investigated by further experiments.

We next turn to understand the model by extracting salient features in the EIS correlated with degradation: Fig. 1c shows the automatic relevance determination (ARD) importance weights of the EIS-Capacity GPR model. Interestingly, the model finds that only two salient frequencies, out of the 120 possibilities in the range of 0.02 Hz–20 kHz, are sufficient to estimate capacity; in Supplementary Fig. 3, we show there is a strong linear change of selected EIS features with cycle number in the Nyquist plot over cycle number at 25 °C. The selected frequencies of 17.80 and 2.16 Hz are located in the low-frequency region, suggesting that it is the change in the interfacial properties that underpins degradation for these batteries; this is consistent with the results obtained in previous works[35], but we demonstrate how a machine-learning framework can aid the interpretation of high-dimensional spectra. When implemented in a battery management system, our EIS-based approach has the potential to enable end-users to know the battery capacity without a full charge–discharge.

**RUL prediction**. One of the ultimate goals of a battery management system is to predict the RUL of a battery and to detect possible hazardous conditions caused by battery aging or abuse. Here, we build a model for RUL prediction from the EIS spectrum (EIS-RUL GPR model).

Figure 2 shows that the EIS-RUL GPR model accurately predicts the RUL of all four testing cells cycled at 25 °C only from

EIS measurements at the current cycle, without requiring EIS measurements from previous cycles. This result suggests that our EIS-machine-learning technology has the potential to be translated into a prototype battery management system.

To further understand the information contained in EIS spectra relative to other electrical signals reported in the literature, we benchmark our method against features extracted from the discharging curve, following recent work[18]. We feed those discharging curve features to the same machine-learning method (GPR model) and using the same training-test split. We observe that our method achieves a lower predictive error (cf. Supplementary Table 1). This suggests that EIS provides significantly richer information about battery health compared to signals that are currently tracked in battery management systems, and those EIS signals can be fruitfully exploited by our GPR method.

**Capacity estimation and RUL prediction at multiple temperatures**. In the context of battery recycling, the problem of battery diagnosis is often more challenging because the historical operating condition of the cell (e.g. temperature) varies all the time. Although temperatures are measured with sensors within a battery module or stack, actual temperatures may deviate considerably due to large temperature gradients under operational conditions. In this section, we explore a simpler toy problem: rather than considering a variation of cycling temperature over time, we ask the question whether the model can still predict RUL, based on the EIS measured at the current cycle, without knowledge of the cycling temperature except that it is constant

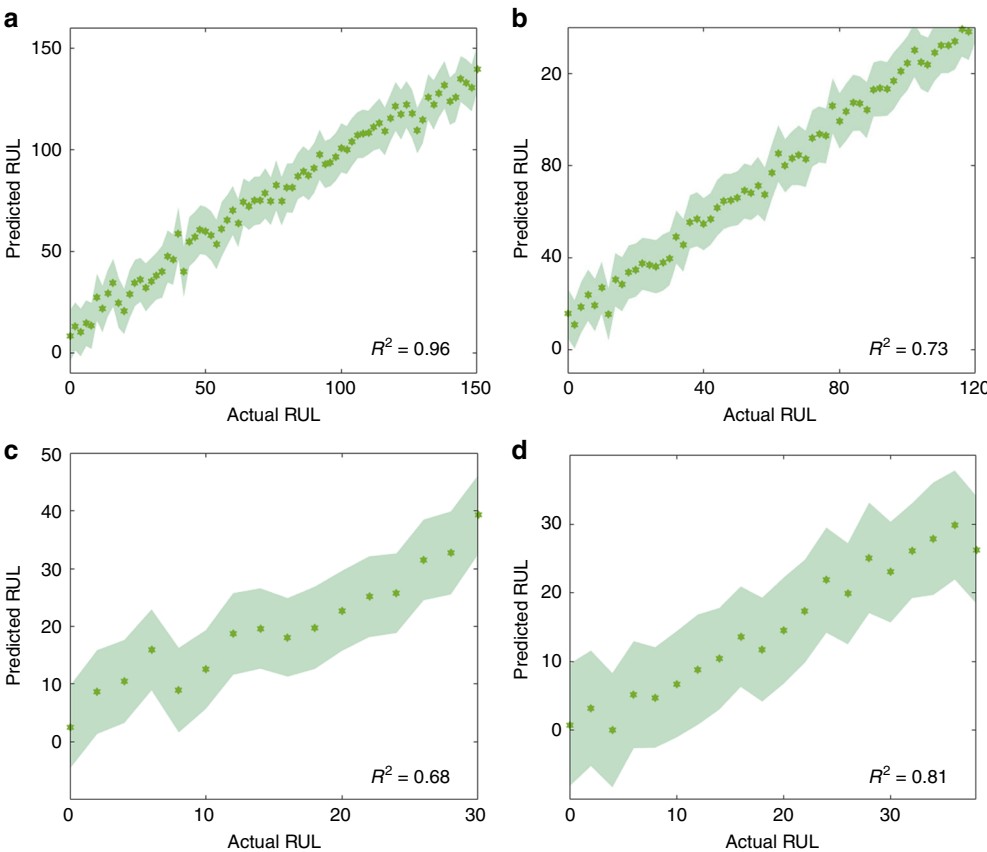

**Fig. 2 Predicting remaining useful life.** The predicted RUL of 25C05–25C08 testing cells **a**–**d** cycled at 25 °C (shown as green curves). The end of life (EoL) of these four testing cells is 150, 120, 30 and 38, respectively, which is defined as the cycle number when the capacity drops below its initial 80%. The testing EIS spectra are collected at the state V (15 min resting after fully charging). The shaded region indicates ±1 standard deviation. $R^2$ is shown on the right bottom in each panel.

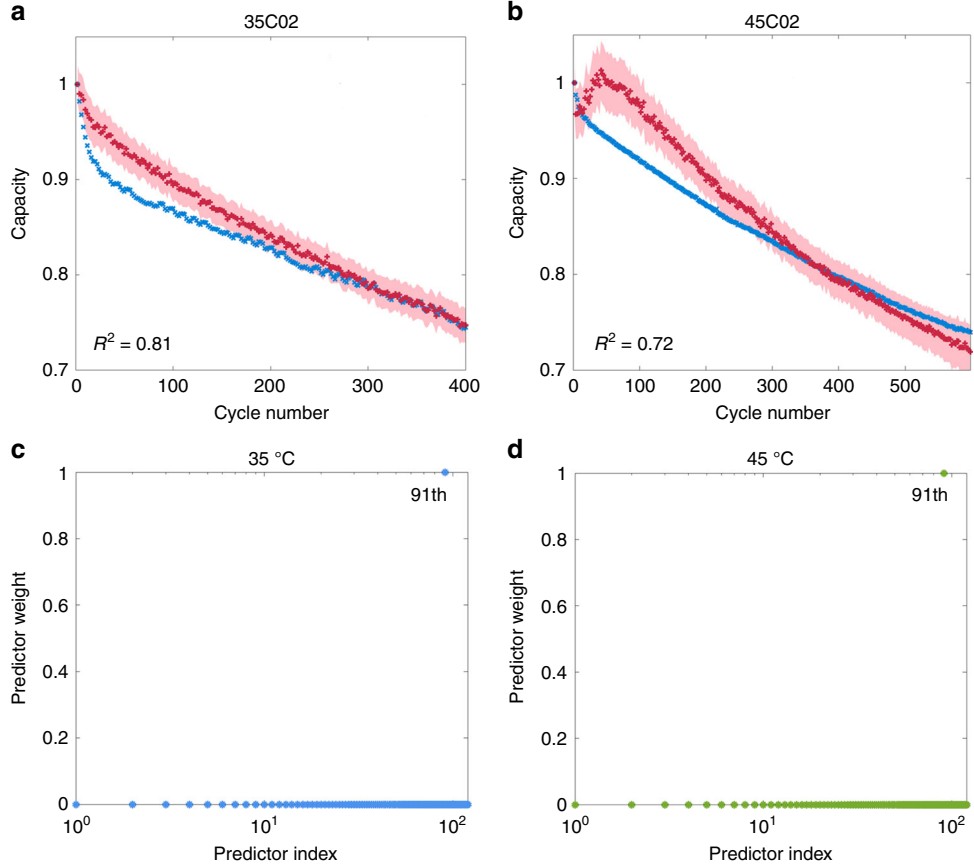

**Fig. 3 Capacity estimation without definite knowledge of the temperature.** The curves show the estimated (red) and measured (blue) capacity for cell cycled at **a** 35 °C and **b** 45 °C; the cycling temperature is not an input to our model. The shaded area indicates ±1 standard deviation. Both EIS spectra are collected at the state V (15 min resting after fully charging). $R^2$ is shown on the left bottom in each panel. ARD shows that the EIS-capacity GPR models at 35 °C **c** and 45 °C **d** assign the largest weight to the 91st feature with the corresponding frequency of 17.80 Hz. The irrelevant features have weights close to zero.

over cycles. We make a further simplification that the temperature is either 25, 35 or 45 °C. We combine the training data acquired at three different temperatures (i.e. 25C01–25C04, 35C01 and 45C01 cells), and in effect forcing GPR to learn features of the EIS that only depends on capacity but not temperature. Figure 3a, b show that our multi-temperature model can estimate capacity of the cells cycled at 35 and 45 °C.

To explore the change of the salient frequency with different temperatures, we apply the ARD method to the EIS-Capacity GPR models for 35 and 45 °C. Figure 3c, d show the ARD importance weights of these two models. Similarly, each model finds that only one salient frequency is sufficient to estimate capacity. The selected frequency, 17.80 Hz, is located in the low-frequency region, consistent with the observation discussed in the previous section.

Following the same idea, we also built a multi-temperature model for RUL prediction. Our EIS-RUL model is able to accurately predict the RUL of cells cycled at three different temperatures (Fig. 4).

## Discussions

In this paper, we show that our GPR models accurately estimate the capacity and predict the RUL using EIS spectra of cells with different degradation patterns cycled at various temperatures but under constant charge/discharge rates. Our method accurately estimates the SoH and RUL of a testing battery cycled at the same charging/discharging rate as the training cells, at any point of its life, from a single impedance measurement, without the

knowledge of the cycling temperature as long as the future operating temperature of a battery is close to its previous operating temperature. Predictions from our model can be attributed back to the impedance spectra, yielding the observation that the low-frequency region of the EIS spectrum is the most predictive.

Our work shows the potential value of signals from EIS in the design of battery management systems. Moreover, we show that GPR with an ARD kernel allows us to identify important features amid many irrelevant ones from high-dimensional measurements. An interesting future direction, stemming from this observation, is that one might not need to perform a full sweep over a broad range of frequencies to obtain signals relevant to degradation. We anticipate that our observation about the value of EIS and GPR can be extended to consider more challenging and realistic settings, such as variations in cycling temperature over time or variations in charge/discharge rate. However, a significantly larger training set is required to cover the different eventualities. We defer consideration of those aspects to future work.

## Methods

**Data generation**. The experiment is carried out by applying a continuous charge–discharge cycle on 12 commercially available 45 mAh Eunicell LR2032 Li-ion coin cells. The cell chemistry is LiCoO$_2$/graphite. The cells are cycled in three climate chambers set to 25 °C (25C01–25C08), 35 °C (35C01 and 35C02) and 45 °C (45C01 and 45C02), respectively. Each cycle consists of a 1C-rate (45mA) CC–CV (constant current–constant voltage) charge up to 4.2 V and a 2C-rate (90 mA) CC (constant current) discharge down to 3 V. EIS is measured at nine different stages of charging/discharging during every even-numbered cycle in the frequency range

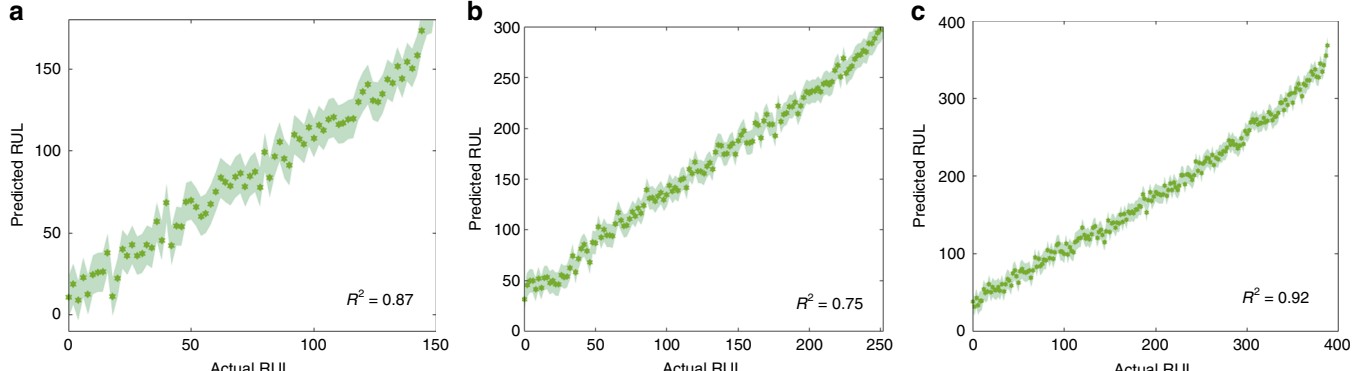

**Fig. 4 Remaining useful life prediction without definite knowledge of the temperature.** Our model accurately predicts the RUL of three testing cells **a–c** cycled at 25, 35 and 45 °C, respectively, without information on the cycling temperature. The EoL of these three testing cells is 150, 252 and 396, respectively, which is defined as the cycle number when the capacity reaches its initial 80%. The testing EIS spectra are collected at state V (15 min resting after fully charging). The shaded region indicates ±1 standard deviation. $R^2$ of our method is shown on the right bottom in each panel.

of 0.02 Hz–20 kHz with an excitation current of 5 mA, following a 15-min open circuit at SoC 0% and SoC 100%. The various conditions of direct current (DC) and relaxation are shown in Supplementary Fig. 1. The loss in capacity is determined after every odd-numbered cycle. EIS and capacity data is available in a public repository.

We use 25C01–25C04, 35C01 and 45C01 cells as the training group, and the others as the testing group. All cells underwent 30 cycles at room temperature of 25 °C before different temperatures were set. The battery is cycled until its end of life (EoL), which is defined as when capacity drops below 80% of its initial value after undergoing these 30 cycles. The capacity retention curves of all cells are shown in Supplementary Fig. 4.

**Gaussian process regression.** To motivate the machine-learning framework, we first consider the problem of estimating capacity from the EIS spectrum. This can be formulated as a regression task: given a training set $\mathcal{D} = \{(\mathbf{x}_i, y_i), i = 1, 2, ..., n\}$ consisting of $n$ pairs of inputs $\mathbf{x}_i$ and outputs $y_i$, compute the predictive distribution of the unknown observations $y^*$ at test indices $\mathbf{x}^*$. We define $\mathbf{X} = [\mathbf{x}_1, ..., \mathbf{x}_n]^\top$ and $\mathbf{Y} = [y_1, ..., y_n]^\top$. In our case the inputs $\mathbf{x}_i = [Z_{re}(\omega_1), Z_{re}(\omega_2), ... Z_{re}(\omega_{60}), ... Z_{im}(\omega_1), Z_{im}(\omega_2), ... Z_{im}(\omega_{60})]^\top$ are the real ($Z_{re}$) and imaginary ($Z_{im}$) parts of impedance spectra collected at 60 different frequencies ($\omega_n, n = 1, 2, ..., 60$) in the range of 0.02 Hz–20 kHz at the current cycle, and the output $y_i$ is the capacity corresponding to the EIS spectrum. The inputs are normalised using the mean and standard deviation of the training data.

GPR performs non-parametric regression with Gaussian processes[36]: We assume that $y_i = f(\mathbf{x}_i + \epsilon_i)$, where $\epsilon_i \sim \mathcal{N}(0, \sigma^2)$ is an independent and identically distributed Gaussian noise. The outputs $\mathbf{f} = (f(\mathbf{x}_1), f(\mathbf{x}_2) \cdots f(\mathbf{x}_N))$ are modelled as a Gaussian random field $\mathbf{f} \sim \mathcal{N}(0, \mathbf{K})$, where $K_{ij} = k(\mathbf{x}_i, \mathbf{x}_j)$ is the covariance kernel. The kernel is a measure of how "close" the points $\mathbf{x}_i$ and $\mathbf{x}_j$ are. The joint distribution of the training set $\{(\mathbf{x}_i, y_i), i = 1, 2, ..., n\}$ and the predicted test output $(\mathbf{x}^*, y^*)$ is

$$\begin{bmatrix} \mathbf{Y} \\ y^* \end{bmatrix} = \mathcal{N}\left(0, \begin{bmatrix} K(\mathbf{X}, \mathbf{X}) + \sigma^2 I & K(\mathbf{X}, \mathbf{x}^*) \\ K(\mathbf{x}^*, \mathbf{X}) & K(\mathbf{x}^*, \mathbf{x}^*) \end{bmatrix}\right) \quad (1)$$

Conditioning on the training set yields the predicted mean on $\mathbf{x}^*$

$$\overline{y}^* = K(\mathbf{x}^*, \mathbf{X})(K(\mathbf{X}, \mathbf{X}) + \sigma^2 I)^{-1}\mathbf{Y}, \quad (2)$$

and its predicted variance

$$\Delta^2 = K(\mathbf{x}^*, \mathbf{x}^*) - K(\mathbf{x}^*, \mathbf{X})(K(\mathbf{X}, \mathbf{X}) + \sigma^2 I)^{-1}K(\mathbf{X}, \mathbf{x}^*) \quad (3)$$

which is a measure of uncertainty.

We implement the EIS-capacity GPR model using the Gaussian processes for machine learning (GPML) toolbox[37] with a zero mean function and a diagonal squared exponential (SE) covariance function with ARD[38]

$$k_{SE}^{ARD}(\mathbf{x}_i, \mathbf{x}_j) = \sigma_f^2 \exp\left[-\frac{1}{2}\sum_{m=1}^{d} \frac{(x_{im} - x_{jm})^T(x_{im} - x_{jm})}{\sigma_m^2}\right] \quad (4)$$

where $\sigma_m$ represents the length scale for feature $m$, $m = 1, 2, ..., d$ and $\sigma_f$ is the signal standard deviation; those hyperparameters are obtained by maximising the marginal likelihood. The ARD covariance function allows the model to downweight and prune irrelevant frequencies from the input by setting $\sigma_m$ to be large. We can interpret the resulting model to understand how important each frequency is: We define the importance of the $m$th frequency as $w_m = \exp(-\sigma_m)$, with $0 < w_m < 1$. The relevant frequencies have large weight values and the irrelevant frequencies have weights close to zero.

In the EIS-RUL GPR model, the input $\mathbf{x}_i$ is the entire EIS spectra, the same as the EIS-capacity GPR model, but the output $y_i$ is the RUL. We use a zero mean function and a linear (LIN) covariance function

$$k_{LIN}(\mathbf{x}_i, \mathbf{x}_j) = \sum_{m=1}^{d} x_{im}^T x_{jm} \quad (5)$$

Although GPR has been used in the literature in the context of Li-ion batteries[17,33,34], we depart from those pioneering works by employing impedance spectra as input, as well employing ARD to shed light on salient frequencies.

## Data availability

Experimental data generated during the study is available in a public repository at https://doi.org/10.5281/zenodo.3633835.

## Code availability

The code is available from the GitHub link at https://github.com/YunweiZhang/ML-identify-battery-degradation.

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

## Acknowledgements

A.A.L., U.S. Y.Z., Q.T. and J.W. acknowledge the funding from the Engineering and Physical Sciences Research Council (EPSRC)—EP/S003053/1.

## Author contributions

A.A.L. and U.S. conceived the study. YW.Z. and Y.Z. analysed the experimental data and developed the ML model. Q.T. and J.W. carried out the experiments. A.A.L. and Y.W.Z. wrote the paper. All authors discussed the results and commented on the manuscript.

## Competing interests
The authors declare no competing interests.
