## [Peer Review File · Nature Communications]

Reviewers' comments:

Reviewer #1 (Remarks to the Author):

The manuscript applies machine learning to a large EIS dataset for commercial lithium-ion batteries to predict cycle number and the remaining useful life (RUL). In summary, this manuscript has some highly significant results that was worthy of publication, but should be revised to be more precise and accurate in the wording of the text. Below are more details comments to this effect, as well as pointing out a few other minor fixes to make.

1. A verb is missing near the beginning of the last sentence on page 2.

2. The manuscript claims that no knowledge of the cycling temperature is needed to predict RUL, which is not true since some assumption must be made about the past and future temperatures, e.g., that they are the same and are constant. RUL is a very strong function of temperature, so any prediction that makes no assumptions on the future temperature will be wildly inaccurate. For example, if a model predicts at cycle 50 that a RUL of 1000 based on data collected at 25 C, then the true RUL would be < 100 if the true future temperature was 55 C.

This issue can be fixed by precisely stating the assumption on the future temperature being implicitly made, for example, that future operating temperatures will be close to the past operating temperatures.

3. The manuscript should more clearly state the limitation of the study that all batteries are charged and discharged in exactly the same way (shown in Figure 1) and exactly the same temperature within each group of batteries (Fig. 4). As such, the machine learned model is shown to be accurate only for batteries that are charged and discharged in exactly the same way and exactly the same temperature in the past, and it is very unlikely that the machine learned model will make good predictions when the charging and discharging profiles are changed such as what happens in a hybrid electric vehicle, or the discharge changes substantially in an electric vehicle, or when the temperature varies.

While I agree that the results are a significant advance worthy of publication in this journal, the manuscript should be more precise and complete in stating the limitations of the study. Given the limitations, many of the claims are too broad as stated, e.g., in the last line of Section V or the

second to last line of page 9. The significant differences between what is shown in the study (perfect charging and discharging profiles for all future and past time, and perfect or nearly perfect temperature at a constant value for all future and past time) means that it is too premature to try to argue that the work "paves the way towards a new paradigm in the design of battery management systems". "Paving the way" is too strong of a statement, a more accurate statement would be "shows the potential value of GPR-based methods in the design of battery management systems"

4. In the first full sentence of text below the Figure 2 caption, insert "for these batteries" between "degradation" and "[37]", as the statement does not hold for all batteries.

5. The anti-heuristics statement at the end of page 9 that goes onto page 10 is an opinion that is not supported by evidence given in this manuscript. In no way do the results of the manuscript show that heuristics is not a valid or even better performing approach. Further, the results do not even compare with heuristics and so any sort of claim cannot be made with regard to the merits or demerits of heuristics.

Furthermore, the relating of heuristic as a way to narrow down features is not a valid comparison to the availability of information-rich data, that is, there is no "either or" that can logically be made between the use of heuristics and the use of information-rich data. For example, no evidence is provided that indicates the proposed approach gives better results than the application of a clever heuristic to the same information-rich dataset. [The example of a valid statement is that EIS contains valuable information on cycle number and RUL prediction.]

Third, what constitutes being "heuristic" is rather subjective, and statements in high-quality journals should not be based on matters of opinion. As an example, some first-principles battery experts would consider all of the methods in the manuscript to be based on "using heuristics to narrow down input features"---exactly the approach being demonized by the authors---because the impedances are trivial examples of features (features are transformations of raw data, the impedance data is a transformation of the raw data, and the manuscript explicitly states that the impedances are features in lines 6 to 7 up from the bottom of page 6), the method proposed in the manuscript narrows down those features (e.g., see Fig. 3), and the approach does not employ first-principles models--arguably making the approach empirical aka heuristic rather than first-principles.

6. As another example of an overstated claim, the introduction states that "Our method accurately estimates the cycle number and RUL of a battery, at any point of its life, from a single impedance spectrum measurement,

without any knowledge of the cycling temperature and SoC", which is not technically correct since some knowledge of the cycling temperature is used in the method, namely, that the temperature of

all cycling in the past and future is exactly (or nearly exactly) the same for the batteries in which estimation of cycle number and RUL is being made.

7. The Conclusions also make too strong of statements, that is, statements are made that are not adequately supported by the results in the manuscript. For example, the first sentence of the Conclusions is stated as being generally true when it is only true within the context of the limitations of the study. The second sentence is actually not correct technically, as the method does use knowledge of the cycling temperature, namely, that the past and future cycling temperature is the same constant value for all cycles.

The comment on interpretability is also too strong. Noting that the low-frequency region is the most predictive is not really an insight, but is rather an observation. An example of insight is that you would be able to confidentially learn something about the underlying mechanisms that reduce the capacity from applying the method. The manuscript cannot claim to be the first to show that identification of frequencies that change in EIS provides insight into battery degradation, since such results have already been established in the literature, e.g., [22] and do not require use of GPR or even machine learning. (Note that the claim that it is "challenging to pick out quantitative features correlated with degradation -- made in the introduction -- is overstated since someone could do such picking by just applying lasso or elastic net, which have already been applied to predicting RUL in Li-ion batteries.)

The last sentence of the conclusion is not supported by the manuscript, in that no evidence is provided that the method is broadly applicable, and the claim regarding "unique features" is overstated in that EIS and machine learning methods have been used before. Instead of focusing so much on salespeak, instead write the results in a precise and accurate manner and, if the results are strong enough (and my opinion is that the results are strong enough), then the results can be published while informing the reader in a technically accurate and precise way.

8. Each author should go through the manuscript in some detail and ask which statements are 100% technically accurate and supported by data--as stated--and then revise the sentences that do not satisfy those conditions.

9. Gaussian processes have been applied to lithium-ion battery RUL prediction problems, but could not find the text in the manuscript that explains how the manuscript's use of Gaussian processes relates to the past applications in the literature. I believe that the specific implementation of Gaussian processes as done in the manuscript has not been applied to lithium-ion batteries, but that review of closely related literature on other ways to apply Gaussian processes to lithium-ion

batteries is needed for the readers (and reviewers) to make a precise understanding of the differences in the applications.

9. The term "magic frequencies" on page 6 is misleading. While machine learning methods that do feature selection can often pick a small number of features that are able to be used to make predictions, typically the features are very highly correlated, in which the features selected by the machine learning method are not necessary significantly better than other features. Frequency data such as in the manuscript are especially highly correlated, so this point applies. When applied to frequency data, typically a model built on taking frequencies close to the selected frequencies gives prediction accuracies that are indistinguishable from the originally selected features -- when applied to a rigorous cross validation (e.g., nested). As such, there is nothing "magic" about the originally selected features.

The paragraph at the end of page 6 that goes onto page 7 refers to interpretability being enabled by the method, but the evidence that is provided is not convincing. Someone could make the same observation on mechanism made in the first two lines of text below the Fig. 2 capture by a quick inspection of the EIS curves as a function of cycle number.

Reviewer #2 (Remarks to the Author):

This paper makes a good-faith attempt to adopt machine learning based interpretation of EIS data for prediction of remaining useful life (RUL) in Li-ion batteries. While the effort is duly noted, the following shortcomings preclude further consideration for publication of this work in this high-impact journal.

- Failure of lithium-ion batteries is a non-linear process and makes prediction of battery life and state of health a difficult task. In this study, there is only one set of data being used as a training set at a given condition. This makes the method more of a fitting than machine learning. This seems to be the case that the authors have just one cell for training and one cell for prediction at 3 different temperatures. Even at same cycling conditions, cells tend to fail at a different cycle number due to the inherent non-linearity of the degradation processes in the system. So just using two cells for a condition with one as a training set and other for prediction can at best be a glorified curve fitting than machine learning. Not to mention, the authors try to identify the cycle number rather than the capacity retention.

- It is not clear why choosing the cycle number as the appropriate output predictor rather than the capacity retention or the state of health. Does this study assume that the failure of all lithium-ion batteries can be similar at a given condition? If so justify the assumption. Elaborate on the logic for choosing cycle number as the output to be predicted.

- Although the coefficients of fit are high enough, it needs to be noted that not enough datasets are used to realistically calculate the coefficient. So even if the model is really good, the study does not have enough data to validate the same.

- The capacity retention curves of the training set as well as the predictor set should be provided (e.g., in the supplementary information).

Reviewer #1

The manuscript applies machine learning to a large EIS dataset for commercial lithium-ion batteries to predict cycle number and the remaining useful life (RUL). In summary, this manuscript has some highly significant results that was worthy of publication, but should be revised to be more precise and accurate in the wording of the text. Below are more details comments to this effect, as well as pointing out a few other minor fixes to make.

We are glad that the referee agrees that our manuscript contains significant results worthy of publication in Nature Communications. We thank the referee for the constructive comments and helpful suggestions and have taken them on board in our revised manuscript.

1. A verb is missing near the beginning of the last sentence on page 2.

We thank the referee for pointing this out. We have added the verb “*interpreted*” in the sentence.

2. The manuscript claims that no knowledge of the cycling temperature is needed to predict RUL, which is not true since some assumption must be made about the past and future temperatures, e.g., that they are the same and are constant. RUL is a very strong function of temperature, so any prediction that makes no assumptions on the future temperature will be wildly inaccurate. For example, if a model predicts at cycle 50 that a RUL of 1000 based on data collected at 25 C, then the true RUL would be < 100 if the true future temperature was 55 C.

This issue can be fixed by precisely stating the assumption on the future temperature being implicitly made, for example, that future operating temperatures will be close to the past operating temperatures.

In the revised manuscript we made an explicit statement that the temperature is fixed when we introduced the model (Section 4).

Moreover, in the revised conclusion, we also state “Our method accurately estimates the SoH and RUL ... without the knowledge of the cycle temperature as long as the future operating temperature of a battery is close to its previous operating temperature”.

3. The manuscript should more clearly state the limitation of the study that all batteries are charged and discharged in exactly the same way (shown in Figure 1) and exactly the same temperature within each group of batteries (Fig. 4). As such, the machine learned model is shown to be accurate only for batteries that are charged and discharged in exactly the same way and exactly the same temperature in the past, and it is very unlikely that the machine learned model will make good predictions when the charging and discharging profiles are changed such as what happens in a hybrid electric vehicle, or the discharge changes substantially in an electric vehicle, or when the temperature varies.

We thank the referee for pointing this out. As the referee suggested, in the revised manuscript we have clearly stated this limitation in the discussion around Figure 1 in the main manuscript. And we also rephrase the sentence in the conclusion to avoid an overstated claim: “Our method accurately estimates the SoH and RUL of a testing

battery cycled at the same charging/discharging rate as the training cells, ...”

While I agree that the results are a significant advance worthy of publication in this journal, the manuscript should be more precise and complete in stating the limitations of the study. Given the limitations, many of the claims are too broad as stated, e.g., in the last line of Section V or the second to last line of page 9. The significant differences between what is shown in the study (perfect charging and discharging profiles for all future and past time, and perfect or nearly perfect temperature at a constant value for all future and past time) means that it is too premature to try to argue that the work "paves the way towards a new paradigm in the design of battery management systems". "Paving the way" is too strong of a statement, a more accurate statement would be "shows the potential value of GPR-based methods in the design of battery management systems"

We have rephrased the claims in the manuscript to reflect the limitations in this study, removing phrases such as “paving the way”. e.g., “...our EIS-machine learning technology has the potential to be translated into a prototype battery management system” (last line of Section V) and “Our work shows the potential value of signals from electrochemical impedance spectroscopy in the design of battery management systems” (Conclusion).

4. In the first full sentence of text below the Figure 2 caption, insert "for these batteries" between "degradation" and "[37]", as the statement does not hold for all batteries.

We thank the referee for pointing this out. We have added the term “for these batteries” in that sentence.

5. The anti-heuristics statement at the end of page 9 that goes onto page 10 is an opinion that is not supported by evidence given in this manuscript. In no way do the results of the manuscript show that heuristics is not a valid or even better performing approach. Further, the results do not even compare with heuristics and so any sort of claim cannot be made with regard to the merits or demerits of heuristics.

Furthermore, the relating of heuristic as a way to narrow down features is not a valid comparison to the availability of information-rich data, that is, there is no "either or" that can logically be made between the use of heuristics and the use of information-rich data. For example, no evidence is provided that indicates the proposed approach gives better results than the application of a clever heuristic to the same information-rich dataset. [The example of a valid statement is that EIS contains valuable information on cycle number and RUL prediction.]

Third, what constitutes being "heuristic" is rather subjective, and statements in high-quality journals should not be based on matters of opinion. As an example, some first-principles battery experts would consider all of the methods in the manuscript to be based on "using heuristics to narrow down input features"---exactly the approach being demonized by the authors---because the impedances are trivial examples of features (features are transformations of raw data, the impedance data is a transformation of the raw data, and the manuscript explicitly states that the impedances are features in lines 6 to 7 up from the bottom of page 6), the method proposed in the manuscript narrows down those features (e.g., see Fig. 3), and the approach does not employ first-principles models--arguably making the approach

empirical aka heuristic rather than first-principles.

We thank the referee for pointing this out. We do not intend to suggest that heuristics are inferior, and we agree that the statements in the previous version were erroneous.

In the revised manuscript, we have rewritten the paragraph to read: “Our work shows the potential value of signals from electrochemical impedance spectroscopy in the design of battery management systems. Moreover, we show that Gaussian Process Regression allows us to extract interpretable information from high-dimensional measurements amid many irrelevant features.” We have also removed any erroneous implied comparison between heuristics and EIS.

6. As another example of an overstated claim, the introduction states that "Our method accurately estimates the cycle number and RUL of a battery, at any point of its life, from a single impedance spectrum measurement, without any knowledge of the cycling temperature and SoC", which is not technically correct since some knowledge of the cycling temperature is used in the method, namely, that the temperature of all cycling in the past and future is exactly (or nearly exactly) the same for the batteries in which estimation of cycle number and RUL is being made.

We have removed and modified this claim in the revised manuscript to be explicit that we assume that the cycling temperature does not change over time. Please see our response to point 2.

7. The Conclusions also make too strong of statements, that is, statements are made that are not adequately supported by the results in the manuscript. For example, the first sentence of the Conclusions is stated as being generally true when it is only true within the context of the limitations of the study. The second sentence is actually not correct technically, as the method does use knowledge of the cycling temperature, namely, that the past and future cycling temperature is the same constant value for all cycles.

We agree with the referee’s concerns. We have substantially rewritten the conclusion. We have significantly modified our claims pertaining to temperature throughout the revised manuscript.

8. The comment on interpretability is also too strong. Noting that the low-frequency region is the most predictive is not really an insight, but is rather an observation. An example of insight is that you would be able to confidentially learn something about the underlying mechanisms that reduce the capacity from applying the method. The manuscript cannot claim to be the first to show that identification of frequencies that change in EIS provides insight into battery degradation, since such results have already been established in the literature, e.g., [22] and do not require use of GPR or even machine learning. (Note that the claim that it is "challenging to pick out quantitative features correlated with degradation -- made in the introduction -- is overstated since someone could do such picking by just applying lasso or elastic net, which have already been applied to predicting RUL in Li-ion batteries.)

In the revised manuscript, we have removed references to “interpretability” and “insights”. The sentence now reads: “Predictions from our model can be attributed back to the impedance spectra, yielding the observation that the low frequency region of the EIS spectrum is the most predictive.”

9. *The last sentence of the conclusion is not supported by the manuscript, in that no evidence is provided that the method is broadly applicable, and the claim regarding "unique features" is overstated in that EIS and machine learning methods have been used before. Instead of focusing so much on salespeak, instead write the results in a precise and accurate manner and, if the results are strong enough (and my opinion is that the results are strong enough), then the results can be published while informing the reader in a technically accurate and precise way.*

In the revised manuscript, we have removed the last sentence. We have also substantially rewritten the conclusion, fully taking on board the referee's advice.

10. *Each author should go through the manuscript in some detail and ask which statements are 100% technically accurate and supported by data--as stated--and then revise the sentences that do not satisfy those conditions.*

We are very grateful to the referee for this suggestion.

11. *Gaussian processes have been applied to lithium-ion battery RUL prediction problems, but could not find the text in the manuscript that explains how the manuscript's use of Gaussian processes relates to the past applications in the literature. I believe that the specific implementation of Gaussian processes as done in the manuscript has not been applied to lithium-ion batteries, but that review of closely related literature on other ways to apply Gaussian processes to lithium-ion batteries is needed for the readers (and reviewers) to make a precise understanding of the differences in the applications.*

We have added more discussion on the application of GP method in the literature in the fourth paragraph in the introduction as well as in the end of Section III, when we introduce the GP method.

12. *The term "magic frequencies" on page 6 is misleading. While machine learning methods that do feature selection can often pick a small number of features that are able to be used to make predictions, typically the features are very highly correlated, in which the features selected by the machine learning method are not necessary significantly better than other features. Frequency data such as in the manuscript are especially highly correlated, so this point applies. When applied to frequency data, typically a model built on taking frequencies close to the selected frequencies gives prediction accuracies that are indistinguishable from the originally selected features -- when applied to a rigorous cross validation (e.g., nested). As such, there is nothing "magic" about the originally selected features.*

We have removed the word "magic" and significantly rephrased the discussion around the salient frequencies that our method identifies. We think the more important conclusion is that the automatic relevance determination method identifies low frequency regions as salient.

13. *The paragraph at the end of page 6 that goes onto page 7 refers to interpretability being enabled by the method, but the evidence that is provided is not convincing. Someone could make the same observation on mechanism made in the first two lines of text below the Fig. 2 capture by a quick inspection of the EIS curves as a function of cycle number.*

Manual inspection could lead to the same conclusion, but we believe there is still merit to automatic relevance determination because: (1) it reassures us that the model is not picking up irrelevant biases, (2) manual inspection that one feature correlates with the output does not necessarily rule out the possibility that multiple features jointly provide more information.

However, in order to avoid any overstated claim on automatic relevance determination results, we have carefully rephrased this part by removing “interpretability”. Also, in the revised manuscript, we stated that our automatic relevance determination result “is consistent with the results obtained in previous works [38], but we demonstrate how a machine learning framework can aid the interpretation of high dimensional spectra” in the last paragraph in Section IV.

Reviewer #2

This paper makes a good-faith attempt to adopt machine learning based interpretation of EIS data for prediction of remaining useful life (RUL) in Li-ion batteries. While the effort is duly noted, the following shortcomings preclude further consideration for publication of this work in this high-impact journal.

1. Failure of lithium-ion batteries is a non-linear process and makes prediction of battery life and state of health a difficult task. In this study, there is only one set of data being used as a training set at a given condition. This makes the method more of a fitting than machine learning. This seems to be the case that the authors have just one cell for training and one cell for prediction at 3 different temperatures. Even at same cycling conditions, cells tend to fail at a different cycle number due to the inherent non-linearity of the degradation processes in the system. So just using two cells for a condition with one as a training set and other for prediction can at best be a glorified curve fitting than machine learning. Not to mention, the authors try to identify the cycle number rather than the capacity retention.

We thank the referee's constructive suggestion. In the revised manuscript, we have performed experiments on significantly more batteries. This increases the data size (over 20,000 EIS spectra) and ascertains the significance of our results. We have also taken on board the referee's concern to focus on capacity rather than cycle number by building the EIS-Capacity models.

We have cycled 6 more cells under 25 °C to test the robustness of our EIS-Capacity model. These six cells failed at different cycle numbers, as the referee expected. Their capacity curves during cycling are shown as below and also in Figure S2 in the Supporting Information (SI).

Figure: Capacity curves during cycling

2. It is not clear why choosing the cycle number as the appropriate output predictor rather than the capacity retention or the state of health. Does this study assume that the failure of all lithium-ion batteries can be similar at a given condition? If so justify the assumption. Elaborate on the logic for choosing cycle number as the output to be predicted.

As the referee suggested, rather than identifying the cycle number, we built a GPR model on identifying capacity from EIS. We trained the GPR model on 25C01-25C04 and tested on 25C05-25C08. We show that our model can indeed estimate the capacity

of cells in the test set with different lifespans. We have shown all these results in Figures 1 and 3 of the revised manuscript.

3. Although the coefficients of fit are high enough, it needs to be noted that not enough datasets are used to realistically calculate the coefficient. So even if the model is really good, the study does not have enough data to validate the same.

We have collected much more data to validate our method in the revised manuscript. Please see our response to Points 1 and 2.

4. The capacity retention curves of the training set as well as the predictor set should be provided (e.g., in the supplementary information).

As the referee suggested, we have added the capacity retention curves of all cells (including training and testing sets) in Figure S2 in the SI.

REVIEWERS' COMMENTS:

Reviewer #1 (Remarks to the Author):

The review comments are adequately addressed.

Reviewer #2 (Remarks to the Author):

The authors have done due diligence in addressing the comments and with an expanded training data set. The revised manuscript is in good standing.